# Phase Transformation and Performance of Mg-Based Hydrogen Storage Material by Adding ZnO Nanoparticles

**DOI:** 10.3390/nano13081321

**Published:** 2023-04-09

**Authors:** Bing Zhang, Ronghan Liu, Hideo Kimura, Yuming Dou, Ziyin Dai, Lirong Xiao, Cui Ni, Chuanxin Hou, Xueqin Sun, Ronghai Yu, Wei Du, Xiubo Xie

**Affiliations:** 1School of Environmental and Material Engineering, Yantai University, No. 30 Qingquan Road, Yantai 264005, China; 2Key Laboratory of Aerospace Materials and Performance (Ministry of Education), School of Materials Science and Engineering, Beihang University, Beijing 100191, China

**Keywords:** hydrogen storage material, ZnO nanoparticle, catalyst, phase change

## Abstract

ZnO nanoparticles in a spherical-like structure were synthesized via filtration and calcination methods, and different amounts of ZnO nanoparticles were added to MgH_2_ via ball milling. The SEM images revealed that the size of the composites was about 2 μm. The composites of different states were composed of large particles with small particles covering them. After the absorption and desorption cycle, the phase of composites changed. The MgH_2_-2.5 wt% ZnO composite reveals excellent performance among the three samples. The results show that the MgH_2_-2.5 wt% ZnO sample can swiftly absorb 3.77 wt% H_2_ in 20 min at 523 K and even at 473 K for 1 h can absorb 1.91 wt% H_2_. Meanwhile, the sample of MgH_2_-2.5 wt% ZnO can release 5.05 wt% H_2_ at 573 K within 30 min. Furthermore, the activation energies (E_a_) of hydrogen absorption and desorption of the MgH_2_-2.5 wt% ZnO composite are 72.00 and 107.58 KJ/mol H_2_, respectively. This work reveals that the phase changes and the catalytic action of MgH_2_ in the cycle after the addition of ZnO, and the facile synthesis of the ZnO can provide direction for the better synthesis of catalyst materials.

## 1. Introduction

In recent decades, the overuse of fossil resources and the waste gas produced by fossil resource combustion have led to the problems of energy dilemma and environmental pollution. In order to solve these problems, people urgently need to find clean energy to replace fossil resources. Among the many clean energy sources that have been discovered, hydrogen energy is favored by researchers, owing to its advantages of being clean, pollution-free, and so on [1,2]. For the applications of the source of hydrogen, hydrogen storage plays a connecting role. Compared with traditional high-pressure gas hydrogen storage and low-temperature liquid hydrogen storage, solid hydrogen storage materials have been widely studied because of their advantages of high safety, small occupation area, low energy consumption, and high hydrogen storage density. For example, rare earth hydrogen storage materials [3,4,5], titanium hydrogen storage materials [6,7,8], vanadium hydrogen storage materials [9,10], magnesium hydrogen storage materials [11,12,13,14,15], etc. MgH_2_ is considered one of the best hydrogen storage materials owing to its abundant reserves, high calorific value (1.43×10^8^ J/kg), high theoretical hydrogen capacity (7.6 wt% H_2_), no pollution, and excellent reversibility. However, the shortcomings of high temperature and slow kinetics in the cycle severely hinder the practical applications [16,17,18].

The research shows that domestic and foreign scholars mainly solve the above problems by adding catalysts [19,20,21,22], nano-limiting [11,23,24], and alloying [25,26,27]. Among them, catalysts are considered to be one of the most effective ways to ameliorate these shortcomings in the cycle. Recently, many kinds of oxides, including V_2_O_3_, TiO_2_, Cr_2_O_3_, etc., [28,29,30,31,32] have been used as catalytic additives to improve the performance of MgH_2_. Du et al. synthesized V_2_O_3_ supported by reduced graphene oxide (rGO) by impregnation and added rGO-supported V_2_O_3_ to Mg_85_Al_15_ alloy, the activation energies for hydrogen absorption and desorption were reduced to 61.6 and 91.3 KJ/mol, respectively [30]. Song et al. prepared Cr_2_O_3_ using spray conversion and added 10 wt% Cr_2_O_3_ to Mg with ball milling. The Mg-10 wt% Cr_2_O_3_ can quickly absorb 4.57 wt% H_2_ in 5 min and 5.93 wt% H_2_ in 1 h at 593 K and 1.2 MPa hydrogen pressure [30]. Liu et al. synthesized TiO_2_@rGO nanoparticles using the solvothermal method. The TiO_2_ nanoparticles are very uniform due to the presence of ethylene glycol and graphene. The MgH_2_-70TiO_2_@rGO-EG composite starts to release hydrogen at 240 °C and can desorb 6.0 wt% H_2_ within 6 min at 300 °C. At the same time, it can absorb 5.9 wt% H_2_ within 2 min at 200 °C [33]. Chen et al. investigated the addition of ZrO_2_ nano-powder to MgH_2_. The results show that the MgH_2_-ZrO_2_ composite can still absorb 4.0 wt% H_2_ at room temperature [34]. Chuang et al. added Nb_2_O_5_ and MWCNT to MgH_2_ with ball milling. After 1 h of ball milling, the activation energies of hydrogen absorption and desorption of Mg-Nb_2_O_5_ and MWCNT composites were reduced to 12.84 KJ/mol and 102.69 KJ/mol, respectively [35]. The above studies indicate that the addition of oxides can promote the performance of MgH_2_ to some extent. As far as we are aware, the influence of ZnO addition on the performance of absorption and desorption of MgH_2_ was rarely discussed. At the same time, little research has been performed on Mg–Zn. Zhong et al. synthesized Mg(Sn, Zn) solid solution, and hydrogenated the solid solution to produce MgZn_2_ in situ [36]. Deledda et al. added Zn to Mg using the ball milling method and finally generated amorphous Mg_45_Zn_55_, and the influence of amorphous Mg_45_Zn_55_ on hydrogen storage material is not obvious [37]. In addition, Liu et al. prepared Mg–Zn alloy using the method of hydrogen plasma–metal reaction. After hydrogen absorption, the Mg in the Mg–Zn alloy is converted to MgH_2_, and a small portion of the Mg reacted with the Zn to form MgZn_2_. The fine particle size and nano-sized grains of the Mg–Zn particles accelerated the nucleation and hydrogen diffusion in the cycle [38]. The studies show that the beneficial effect on Mg with the increase in the amount of ZnO decreases in the cycle. The phase change during the cycle is also closely related to the pressure of hydrogen. When the pressure of hydrogen absorption is higher than 1 MPa, MgZn_2_ will be generated, and in the process of desorption, the multi-phase Mg–Zn intermetallics are formed [37,38,39]. In this work, we added the different amount of ZnO into MgH_2_ by the most common method of ball milling and investigated the optimal doping amount of ZnO and the phase change in the cycle.

## 2. Materials and Methods

### 2.1. Materials

Triethylamine (TEA), terephthalic acid (PTA), dimethylformamide (DMF), and Zn(NO_3_)_2_·6H_2_O were purchased from Macklin (Shanghai, China) without purification.

### 2.2. Preparation of the ZnO

The preparation process of ZnO nanoparticles is shown in Figure 1. The preparation method is as follows: Firstly, add 150 mL DMF and 3.75 mmol Zn(CH_3_COO)_2_·2H_2_O into the beaker and stir for 30 min, named solution A. Next, add 100 mL DMF, 9 mmol PTA and 3.125 mL TEA and stir for 30 min, named solution B. Then, pour solution B into solution A and continue stirring for 1 h. After stirring, the white powder is obtained by vacuum filtration of the transparent solution. Finally, put the white powder into the tube furnace and heat the sample to 700 °C under air at 5 °C/min, and the ZnO is obtained after being left for 10 h.

### 2.3. Preparation of the MgH_2_-xZnO Nanocomposites (x = 2.5, 5, 7.5 wt%)

The amounts of ZnO particles of 2.5, 5, and 7.5 wt% were added to MgH_2_ powder by the method of ball milling, respectively. Firstly, the powder of ZnO and MgH_2_ was mixed in a grinding tank in the glove boxes filled with Ar. Then, the grinding tank was removed from the glove boxes, after vacuuming, and high purity Ar of 0.2 MPa was added. The process was repeated 1~2 times before the ball milling. The technological parameters in the ball milling process were as follows: the total time of the process of ball milling was 720 min (the ratio of run to stop was 60 min 3:1), the rotation speed was 450 r/min and the ratio of ball to powder was 30:1. Finally, the powder was collected in a glove box filled with Ar to prevent avoid oxidation

### 2.4. Characterization

The XRD data of different states of Mg-xZnO nanocomposite were measured in the 2θ range from 20 to 90°and a scanning rate of 6°/min using monochromatic Cu-K_α_ radiation. The images of morphology and microstructures of the prepared Mg-xZnO nanocomposites were measured with scanning electronic microscopes (SEM, JEOL JSM-7610F, Tokyo, Japan). In order to avoid oxidation, the samples of different states of Mg-xZnO nanocomposite were placed in a mikrouna glove box (C(H_2_O) < 1 mg/L, C(O_2_) < 1 mg/L, China). 

The properties of kinetics and thermodynamics of Mg-xZnO composites at each temperature were performed in Sieverts-type equipment. The powders of 300~500 mg after ball milling were packed into a specially designed steel tube for testing and the samples were activated at least 2 times under 673 K and 4 MPa and under vacuum (≤0.1 Pa). In the isothermal de/hydrogenation experiments, the pressure of hydrogen absorption and desorption was 4 MPa and under vacuum (≤0.1 Pa), respectively, and the test time of hydrogen absorption and desorption was 1 h.

## 3. Results and Discussion

### 3.1. Characterization of ZnO

Figure 2a shows the phase composition of as-prepared ZnO nanoparticles. It shows that the strong diffraction peak at 2θ = 31.77°, 34.42°, 36.25°, 47.54°, 56.60°, 62.86°, 66.38°, 67.96°, 69.10°, 72.56°, 76.95°, 81.37°, and 89.60°, which agree with the standard PDF card of ZnO (PDF#36-1451). Moreover, in Figure 2a, the absence of other phases indicates the high purity of the prepared ZnO nanoparticles. In the SEM image of nanoparticles shown in Figure 2b, it can be seen that the prepared ZnO nanoparticles have a spheroidal structure. The size of ZnO nanoparticles is about 300~500 nm, while the shape is regular and the particle size is homogeneous. To show the composition of the prepared ZnO nanoparticles, the element mapping results show that the O and Zn elements are uniform. Moreover, the O and Zn seem to coincide with each other, suggesting that the nanoparticles are ZnO. The above results all prove the successful preparation of ZnO nanoparticles.

### 3.2. Characterization of MgH_2_-xZnO Composite (x = 2.5, 5, 7.5 wt%)

So as to further determine the effect of ZnO, different contents of ZnO were added to MgH_2_ to explore the phase changes. Figure 3 shows the XRD patterns of MgH_2_-xZnO with different states. The XRD patterns show that the most diffraction peaks can be determined with MgH_2_ (PDF#12-0697), and the diffraction peaks at 31.7°, 34.4°, 36.3°, 47.5°, 56.6°, 62.8, 69.1°, and 76.9° can be ascribed by the ZnO (PDF#36-1451) in Figure 3a. Meanwhile, the diffraction peak is found at 2θ = 42.9°, which is due to the slow oxidation of the composite during the test [40]. Figure 3a also shows that with the increase in the content of ZnO, the diffraction peaks of ZnO becomes higher, and the phenomenon shows that ZnO does not decompose after ball milling. All the above results indicate the successful addition of ZnO. As exhibited in Figure 3b, the diffraction peaks of ZnO and MgH_2_ disappear, and the diffraction peaks of Mg (PDF#12-0697) appear. It is worth noting that some diffraction peaks of uncertainty are detected at 2θ = 38−50°, which is due to the low hydrogen pressure results in Mg reacting with ZnO to form the multi-phase Mg-Zn intermetallics (MgZn, Mg_2_Zn_3,_ and Mg_2_Zn_11_, etc, named Mg_x_Zn_y_) during the process of hydrogen release [38]. In order to have a clearer observation, the above area is enlarged, as shown in Figure 3e. However, the diffraction peak of MgO still exists in Figure 3b, it was caused by oxidation during the test. Figure 3c shows that the XRD patterns of the sample of re-hydrogenation, the conversion of Mg and H_2_ into MgH_2,_ and the diffraction peaks at 2θ = 38−50° disappear. At the same time, the diffraction peaks at 20.8°, 22.3°, 37.3°, 39.9°, 40.5°, 41.3, 42.2°, 45.4°, 46.9°, 51.5°, 53.6°, and 54.7° can be ascribed to the MgZn_2_ (PDF#34-0457) in Figure 3c. In Figure 3d, the diffraction peak of MgZn_2_ can be seen more clearly, which further proves that the intensity of the diffraction peak increases with the increase in the content of the catalyst. Liu et al. prepared Mg–Zn alloy and the XRD results showed that a small amount of MgZn_2_ was generated in the cycling process, and the performance test showed that the Mg–Zn composite had better performance than Mg [38]. Zhong et al. synthesized Mg (Sn, Zn) solid solution matrix. After hydrogen absorption, MgZn_2_ was generated in situ [36]. The superior presence of MgZn_2_ promoted the hydrogen absorption and release process. These results indicate that MgH_2_-ZnO composites can be converted into MgZn_2_ phase and Mg_x_Zn_y_ phase during the cycle process, which has a positive catalytic effect on the subsequent hydrogenation reaction, thus enhancing the kinetics.

In order to study the catalytic effects of ZnO nanoparticles on MgH_2_, Figure 4 displays the kinetics curves of hydrogen absorption and desorption of MgH_2_-xZnO. It can be seen that the composite with a different amount of ZnO shows a similar absorption kinetic above 623 K in Figure 4a–c. The hydrogen storage capacity of MgH_2_-2.5 wt% ZnO, MgH_2_-5 wt% ZnO, and MgH_2_-7.5 wt% ZnO were 6.077, 5.994, and 5.957 wt% H_2_ at 673 K, respectively. At the same time, this phenomenon also indicates that the maximum hydrogen content decreases with the increase in ZnO content. However, the decrease is not particularly obvious, which is well demonstrated by TPD test. This is caused by the inability of ZnO to store hydrogen, thus reducing the hydrogen capacity of MgH_2_-xZnO composites. When the sample absorbs hydrogen at high temperatures, the sample will also undergo the hydrogen desorption process. However, at this temperature, the rate of hydrogen absorption is much higher than the rate of hydrogen desorption, and the presence of a catalyst also accelerates the process of hydrogen absorption. The hydrogen absorption process belongs to the gas–solid reaction, which will eventually reach the equilibrium state. Due to the existence of the catalyst, the hydrogen absorption rate is too fast, leading to the increase in hydrogen absorption, but it is not in the equilibrium state at this time. After a period of time, the equilibrium state is reached, resulting in the decrease in the maximum hydrogen absorption. It is worth noting that the MgH_2_-2.5 wt% ZnO sample can absorb 4.70 wt% H_2_ in 1 h at 523 K. However, MgH_2_-5 wt% ZnO and MgH_2_-7.5 wt% ZnO can only absorb 4.26 wt% H_2_ and 2.92 wt% H_2_, respectively, under the same conditions. Meanwhile, Mustafa et al. studied the effect of adding CeO_2_ nano-powder on the hydrogen storage performance of MgH_2_. The study shows that the MgH_2_-5 wt% CeO_2_ composite can absorb about 4.1 wt% H_2_ at 573 K within 1 h. Even at 593 K, the MgH_2_-5 wt% CeO_2_ composite can only absorb approximately 4.4 wt% H_2_ [41], and Du et al. synthesized the Mg_85_Al_15_-V_2_O_3_@rGO composite so it can absorb 5.05 wt% H_2_ under the same conditions [28]. The results show that the hydrogen capacity of the Mg_85_Al_15_-V_2_O_3_@rGO composite is lower than the MgH_2_-2.5 wt% ZnO composite. Even at 473 K, MgH_2_-2.5 wt% ZnO also absorbs 1.91 wt% H_2_ in 1 h. At the same time, the MgH_2_-5 wt% ZnO and MgH_2_-7.5 wt% ZnO can only absorb 1.23 wt% H_2_ and 0.68 wt% H_2_, respectively. However, Liu et al. display that the Mg–Zn alloy can absorb 2.30 wt% H_2_ in 1 h [38]. The results show that the MgH_2_-2.5 wt% ZnO composite is only 0.4 wt% lower. In this work, the MgH_2_-2.5 wt% ZnO composite still exhibits better kinetic performance in the three samples as the temperature drops lower. In terms of hydrogen desorption, dehydrogenation kinetics remain similar at high temperatures. In Figure 4e,f, the MgH_2_-2.5 wt% ZnO, MgH_2_-5 wt% ZnO, and MgH_2_-7.5 wt% ZnO composite can release approximately 5.469, 5.175, and 4.858 wt% H_2_, respectively, under 648 K. The results show that the hydrogen desorption capacity also decreases with the increase in ZnO content. MgH_2_-2.5 wt% ZnO composite showed the best hydrogen desorption performance in the three samples. Meanwhile, Novakovic et al. added VO_2_ to MgH_2_ by ball milling. In the cycle, the metastable α-MgH_2_ phase is transformed into a stable β-MgH_2_ phase and the crystalline VH_2_ phase appears due to the presence of VO_2_. The results show that Mg-15VO_2_ composite can release about 5 wt% H_2_ after 5000 s at 350 °C [42]. The data show that its hydrogen desorption capacity is lower than that of the MgH_2_-2.5 wt% ZnO composite in this study. Moreover, as the decrease in the temperature of hydrogen release, the dehydrogenation kinetics of the three samples are different. The differences between the three samples are as follows: the MgH_2_-2.5 wt% ZnO sample can release 5.47 wt% H_2_ within 1 h at 623 K, and MgH_2_-5 wt% ZnO and MgH_2_-7.5 wt% ZnO can only release 4.90 wt% H_2_ and 4.58 wt% H_2_, respectively, with the same conditions. Ren et al. synthesized the MgH_2_-Ni/Fe_3_O_4_@MIL composite, and the composite was able to release 5.02 wt% H_2_ at 598 K [43]. The capacity is lower than the MgH_2_-2.5 wt% ZnO sample at 623 K in this study. Meanwhile, the above results indicate that the sample of MgH_2_-2.5 wt% ZnO has the best kinetic performance among the three samples.

To further assess the catalysis effects of ZnO on MgH_2_, the E_a_ of MgH_2_-xZnO was calculated by the model of Johnson–Mehl–Avrami–Kolmogorov (JMAK) and the model is investigated by the Equation (1)
ln[−ln(1 − α)] = ŋlnk + ŋlnt(1)
where α is the percent conversion at t, k is a parameter, and η represents the Avrami exponent. Additionally, η and ηlnk can be gained by the slope and intercept of the fitting lines of ln [−ln(1−α)] vs. ln t at different temperatures, and the curves of ln [−ln(1−α)] vs. lnt are shown in Figure 5. At the same time, Table 1 details the results of the fit of the kinetic curves with appropriate standard deviation. Table 1 shows that the standard deviation of the fitted curves of ln [−ln(1−α)] vs. ln t for the measured temperatures of all samples is lower than 0.043, which also indicates that the data are very stable and the accuracy of the results of the fitting curve of ln [−ln(1−α)] vs. lnt at different temperatures were further verified. Next, the E_a_ can be calculated via the Arrhenius equation in Equation (2)

lnk = E_a_/RT + lnA
(2)
where A represents a coefficient, T is the temperature, R is the gas constant (8.314 KJ/mol·K) and the lnk is obtained by the JMAK equation. 

The plots of the lnk vs. 1000/T of hydrogen absorption and desorption are shown in Figure 6a,b. In Figure 6a and Table 1, the value of E_ab_ (the activation energy of hydrogen absorption) of MgH_2_-2.5 wt% ZnO, MgH_2_-5 wt% ZnO, and MgH_2_-7.5 wt% ZnO were calculated to be 72.00, 82.66, and 88.43 KJ/mol, respectively. Singh et al. synthesized different sizes of CeO_2_ nanoparticles with ball milling, and the kinetics effect of MgH_2_-CeO_2_ was studied. The E_ab_ was 84 KJ/mol [28]. The E_a_ for the hydrogenation of the sample of MgH_2_-2.5 wt% ZnO is lower than that of the Mg-TiO_2_@C sample. So as to further study the sample of MgH_2_-2.5 wt% ZnO, the value of E_de_ (the activation energy of hydrogen release) of MgH_2_-2.5 wt% ZnO, MgH_2_-5 wt% ZnO and MgH_2_-7.5 wt% ZnO are calculated to be 107.58, 116.35 and 128.41 KJ/mol, respectively, in Figure 6b and Table 1. In the three samples, the E_ab_ and E_de_ of MgH_2_-2.5 wt% ZnO are the lowest, which also demonstrates the MgH_2_-2.5 wt% ZnO with the best kinetic performance. Research shows that TiO_2_ is an excellent catalyst for Mg-based hydrogen storage material. Zhang et al. synthesized the Mg-10 wt% TiO_2_@C composite, the E_de_ for the composite was 106 ± 4 KJ/mol [44]. All the above results show that MgH_2_-2.5 wt% ZnO composites have the best performance among the three samples, and the activation energy of hydrogen absorption and desorption is superior to other materials.

Figure 6c reveals non-isothermal dehydrogenation curves of MgH_2_ and MgH_2_-xZnO composites. The test results show that the initial dehydrogenation temperatures of MgH_2_-2.5 wt% ZnO, MgH_2_-5 wt% ZnO, and MgH_2_-7.5 wt% ZnO decreased by 72.5, 66.5, and 54.0 °C, respectively, compared to pure MgH_2_. Moreover, the hydrogen capacity of MgH_2_-2.5 wt% ZnO, MgH_2_-5 wt% ZnO and MgH_2_-7.5 wt% ZnO are 6.61, 6.33, and 6.22 wt% H_2_. The capacity of the above three samples is higher than 5.79 wt% H_2_ of pure MgH_2_ and Mg_85_Al_15_-V_2_O_3_@rGO composite [19]. The result of TPD also indicated the lowest dehydrogenation temperature of MgH_2_-2.5 wt% ZnO in Figure 6c. In a word, the above data indicate that MgH_2_-2.5 wt% ZnO has the best performance among the three samples.

### 3.3. Microstructure of MgH_2_-xZnO (x = 2.5, 5, 7.5 wt%)

To study the effect of MgH_2_ by adding ZnO with different amounts, the micromorphology of the MgH_2_-xZnO composites of ball milling after dehydrogenated and re-hydrogenated are characterized by SEM and shown in Figure 7. The SEM images show that the size of the composites is about 2 μm. Figure 7a shows that ZnO nanoparticles were uniformly attached to the surface of MgH_2_. This phenomenon shows that the catalyst can be evenly dispersed on the surface of the MgH_2_ by ball milling.

After dehydrogenation, the SEM exhibit the size of the particle to also be about 2 μm and the similar morphology with the state of ball milling. According to the phase change of XRD, MgH_2_ is transformed into Mg, while small particles on the surface transform from ZnO into Mg_x_Zn_y_, see Figure 7b. Moreover, Figure 7c shows the SEM image of MgH_2_-2.5 wt% ZnO after re-hydrogenation, the results display no significant morphology change from the other two states. According to Figure 3c, the small particles on the surface transform from Mg_x_Zn_y_ into MgZn_2_ in Figure 7c. The morphological evolution of the sample of MgH_2_-5 wt% ZnO and MgH_2_-7.5 wt% ZnO are similar to MgH_2_-2.5 wt% ZnO.

In order to further ensure the element composition of the different states of MgH_2_-ZnO, the element mapping of the composites with different states of ball milling, dehydrogenation, and re-hydrogenation. Figure 8 shows that the elements of Mg, Zn, and O are uniformly distributed, and Mg is the dominant element. The XRD pattern shown in Figure 3a also confirms that ZnO exists in the composite after ball milling, implying that the ZnO remains stable during ball milling. As for MgO, it is inevitable for the formation of MgO for the high activity of the powder after ball milling; therefore, the O element in Figure 8 is also contributed from MgO to some extent. 

The element mapping of dehydrogenated MgH_2_-5 wt% ZnO displays the elements of Mg and Zn uniformly distributed which prove the phase change, see Figure 9. The elements mapping of the sample of re-hydrogenated is shown in Figure 10. 

The elements of Mg and Zn also uniformly distributed, and it is proven that the Mg_x_Zn_y_ turns into MgZn_2_. It can be seen that after the addition of ZnO, the catalyst does not decompose. The phase of the catalyst will be transformed when the sample changes to the hydrogen absorption or hydrogen desorption state.

## 4. Conclusions

In this work, we synthesize a spheroidal structure ZnO nanoparticle via filtration and calcination methods, the nanoparticle size varied from 300 to 500 nm. Different amounts of ZnO were introduced to the MgH_2_ through the ball milling method. After ball milling, the ZnO is not decomposed and is uniformly dispersed on the surface of MgH_2_. After the desorption process, the ZnO reacts with Mg to form Mg_x_Zn_y_ due to the low pressure of hydrogen. After the absorption, the Mg in the Mg–Zn alloy is converted to MgH_2_, and a small portion of the Mg reacted with the Zn to form MgZn_2_. After the performance test, the MgH_2_-2.5 wt% ZnO composite has the best kinetic performance. Compared with pure MgH_2_, the initial dehydrogenation temperature of MgH_2_-2.5 wt% ZnO is 40 °C lower than it. As for isothermal tests, MgH_2_-2.5 wt% ZnO composite can release about 5.47 wt% at 623 K and can absorb 4.70 wt% H_2_ in 1 h at 523 K. However, MgH_2_-5 wt% ZnO and MgH_2_-7.5 wt% ZnO can releases/absorb 4.90 wt%/4.26 wt% H_2_ and 4.58 wt%/2.92 wt% H_2_ with the same condition, respectively. According to the calculation of activation energy, the E_ab_ and E_de_ of MgH_2_-2.5 wt% ZnO are the lowest, which are 72.00 KJ/mol and 107.58 KJ/mol, respectively. Although the addition of ZnO plays a certain catalytic role, the catalytic effect is not excellent compared with other kinds of catalysts.

## Figures and Tables

**Figure 1 nanomaterials-13-01321-f001:**
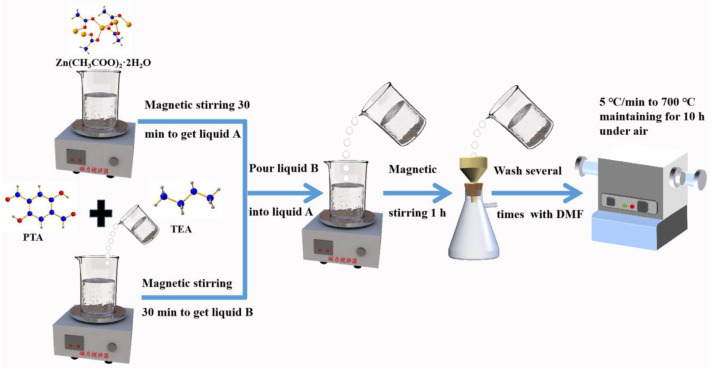
Schematic illustration for synthesizing ZnO nanoparticles.

**Figure 2 nanomaterials-13-01321-f002:**
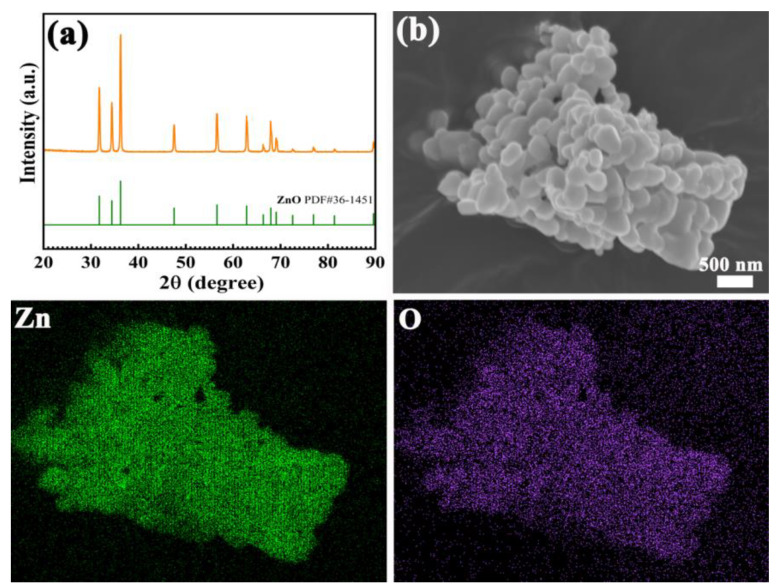
XRD patterns (**a**); SEM (**b**) and element mapping images of ZnO nanoparticles.

**Figure 3 nanomaterials-13-01321-f003:**
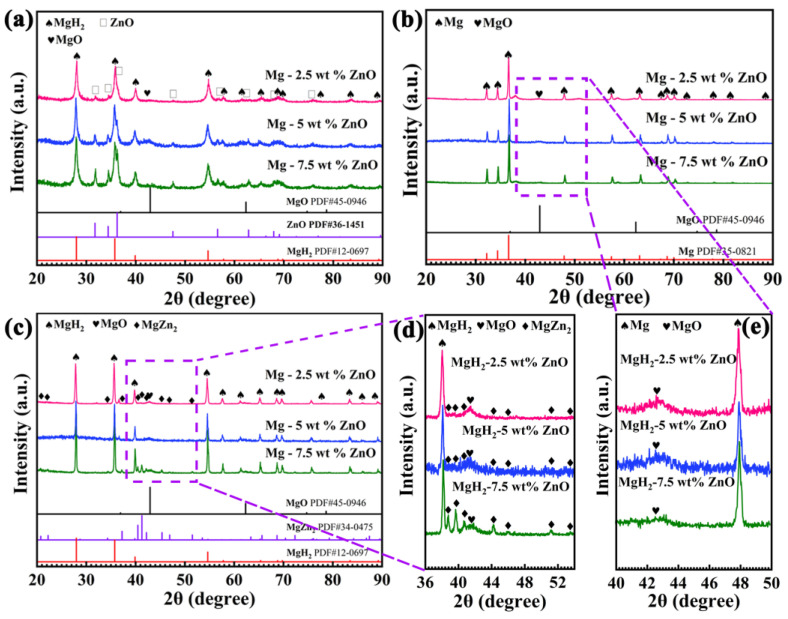
XRD patterns of MgH_2_-xZnO with different sates: ball milling (**a**); after hydrogen release (**b**); and after re-hydrogen (**c**); amplification diagram of re-hydrogenation (**d**); and amplification diagram of dehydrogenation (**e**).

**Figure 4 nanomaterials-13-01321-f004:**
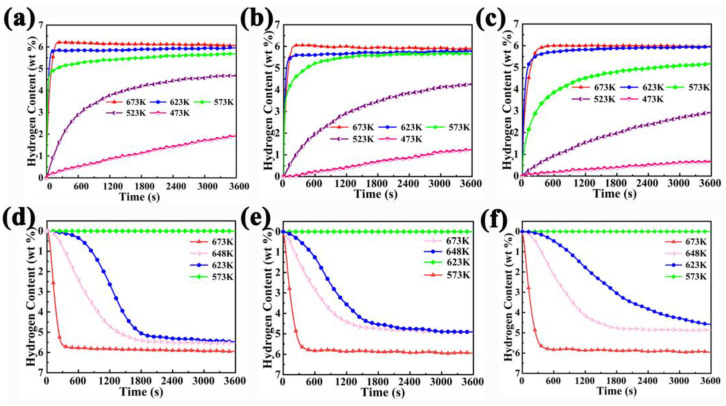
The kinetics curves of hydrogen absorption for MgH_2_-2.5 wt% ZnO (**a**); MgH_2_-5 wt% ZnO (**b**); MgH_2_-7.5 wt% ZnO (**c**); and the kinetics curves of hydrogen desorption for MgH_2_-2.5 wt% ZnO (**d**); MgH_2_-5 wt% ZnO (**e**); MgH_2_-7.5 wt% ZnO (**f**) at different temperatures.

**Figure 5 nanomaterials-13-01321-f005:**
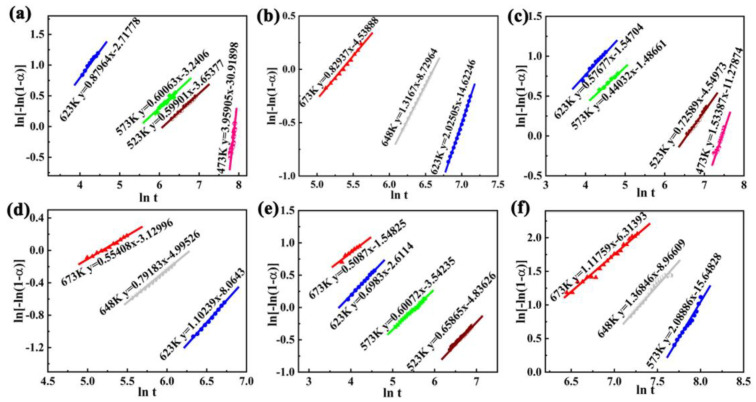
The ln [ln(1−α)] vs. lnt plots for hydrogenated and dehydrogenated of MgH_2_-2.5 wt% ZnO (**a**,**b**); MgH_2_-5.0 wt% ZnO (**c**,**d**); and MgH_2_-7.5 wt% ZnO (**e**,**f**) nanocomposites.

**Figure 6 nanomaterials-13-01321-f006:**
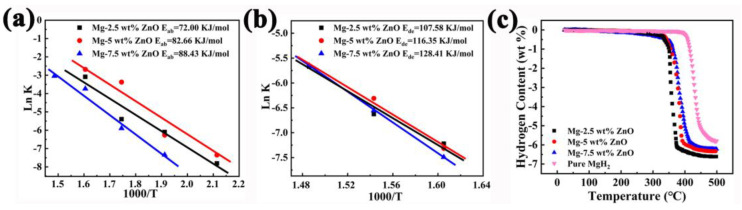
The lnk vs. 1000/T plots of re-hydrogenated (**a**,**b**) dehydrogenated MgH_2_-xZnO, and the comparison of TPD curves (**c**).

**Figure 7 nanomaterials-13-01321-f007:**
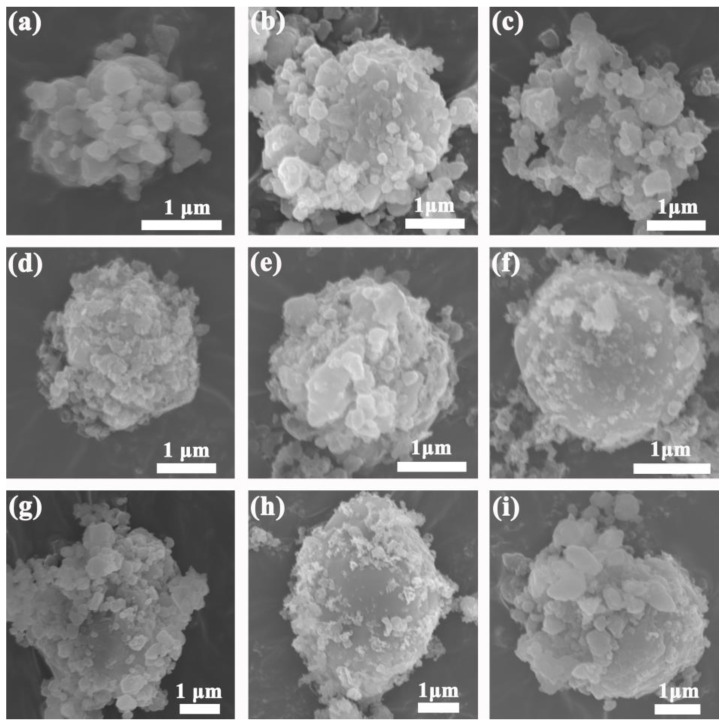
The SEM images of the state of ball milling (**a**); after hydrogen release (**b**); and after re-hydrogen (**c**) of MgH_2_-2.5 wt% ZnO; the images of SEM of the state of ball milling (**d**); after hydrogen release (**e**); and after re-hydrogen (**f**) of MgH_2_-5 wt% ZnO; the images of SEM of the state of ball milling (**g**); after hydrogen release (**h**); and after re-hydrogen (**i**) of MgH_2_-7.5 wt% ZnO.

**Figure 8 nanomaterials-13-01321-f008:**
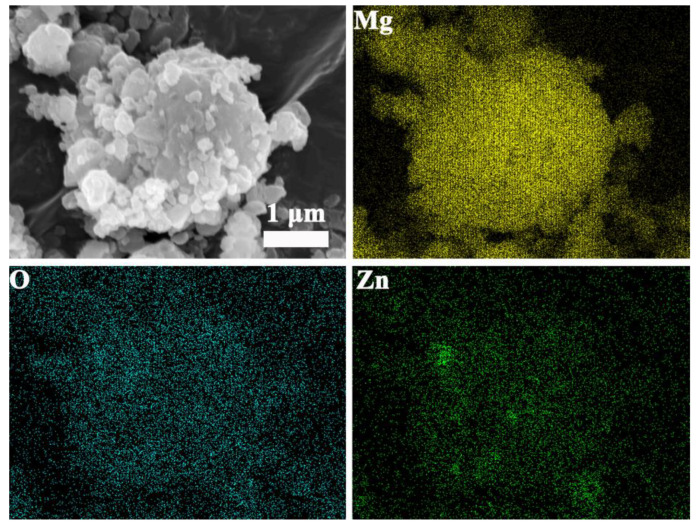
The element mapping images of MgH_2_-5 wt% ZnO composite of ball milling.

**Figure 9 nanomaterials-13-01321-f009:**
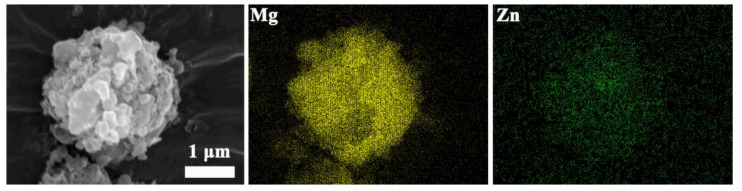
The element mapping images of MgH_2_-5 wt% ZnO composite after dehydrogenation.

**Figure 10 nanomaterials-13-01321-f010:**
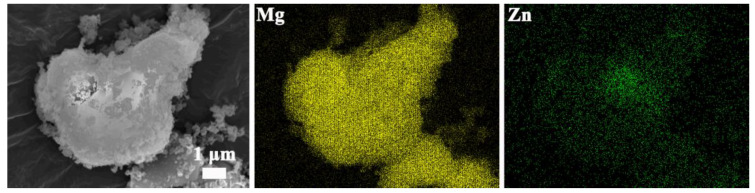
The element mapping images of MgH_2_-5 wt% ZnO composite after re-hydrogenation.

**Table 1 nanomaterials-13-01321-t001:** The hydrogenation of the ln [−ln(1 − α)] vs. lnt equation, the standard deviation and activation energy of Mg-2.5 wt% ZnO, Mg-5.0 wt% ZnO and Mg-7.5 wt% ZnO nanocomposites.

	The Hydrogenation of the ln [−ln(1−α)] vs. lnt Equation and the Standard Deviation		The Dehydrogenation of the ln [−ln(1−α)] vs. lnt Equation and the Standard Deviation	
MgH_2_-2.5 wt% ZnO	623 K: y = 0.87964x − 2.71778 573 K: y = 0.60063x − 3.2406 523 K: y = 0.59901x − 3.65377 473 K: y = 3.95905x − 30.91898	SD = 0.01835 SD = 0.00538 SD = 0.00183 SD = 0.07303	673 K: y = 0.82937x − 4.53888 648 K: y = 1.3167x − 8.72964 623 K: y = 2.02505x − 14.62246	SD = 0.01383 SD = 0.00268 SD = 0.01026
	E_ab_ = 72.00 KJ/mol		E_de_ = 107.58 KJ/mol	
MgH_2_−5.0 wt% ZnO	623 K: y = 0.57677x − 1.54704 573 K: y = 0.44032x − 1.48661 523 K: y = 0.72589x − 4.54973 473 K: y = 1.53387x−11.27874	SD = 0.01784 SD = 0.00950 SD = 0.01053 SD = 0.03496	673 K: y = 0.55408x − 3.12996 648 K: y = 0.79183x − 4.99526 623 K: y = 1.10239x − 8.0643	SD = 0.00978 SD = 0.00451 SD = 0.01028
	E_ab_ = 82.66 KJ/mol		E_de_ = 116.35 KJ/mol	
MgH_2_−7.5 wt% ZnO	673 K: y = 0.5087x − 1.54825 623 K: y = 0.6983x − 2.6114 573 K: y = 0.60072x − 3.54235 523 K: y = 0.65865x − 4.83626	SD = 0.02310 SD = 0.00728 SD = 0.00976 SD = 0.00971	673 K: y = 1.11759x − 6.31393 648 K: y = 1.36846x − 8.96609 623 K: y = 2.08886x − 15.64828	SD = 0.03828 SD = 0.04221 SD = 0.03016
	E_ab_ = 88.43 KJ/mol		E_de_ = 128.41 KJ/mol	

## Data Availability

The data presented in this study are available on request from the corresponding author.

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
