# Peer review of "Phase Transformation and Performance of Mg-Based Hydrogen Storage Material by Adding ZnO Nanoparticles"

_nanomaterials, 2023, doi:10.3390/nano13081321_

Round 1

Reviewer 1 Report

The authors of the manuscript “Phase transformation and performance of Mg based hydrogen storage material by adding ZnO Nanoparticles” report on hydrogen sorption/desorption properties of MgH2/ZnO systems with 2.5%, 5%, 7.5% of ZnO content (wt%).

The ZnO is self-prepared and the composite obtained via ball milling. The samples are characterized by XRD, SEM, Sieverts-type apparatus. Experiments seem to be carefully done and it is concluded that 2.5 wt%  of ZnO is the best catalyst. Kinetic properties and activation energies are determined.

There are some unclear points:

-          Fig. 4: molar mass of three different composition varies, how is it possible that maximum hydrogen content is always the same (~6wt%) and why is decreases with time (@673K) for some samples?

-          There is ZnO and after the first re-hydrogenation ZnO is not present (Fig. 3), MgO seems to be present there. In Fig. 8 they claim no MgO. The question is – where is oxygen? Or what means the statement “It can be see that after the addition of ZnO, the catalyst not decomposes.”?

-          Since ZnO is gone after first hydrogenation/dehydrogenation – what is the catalytic action of their composite for subsequent hydrogenation events?

The language shall be double checked, and strange statements (The Ede, paper exhibite and many others)  shall be corrected.

Once points raised above are clarified the manuscript can be subject to publication.

Author Response

The authors of the manuscript “Phase transformation and performance of Mg based hydrogen storage material by adding ZnO Nanoparticles” report on hydrogen sorption/desorption properties of MgH2/ZnO systems with 2.5%, 5%, 7.5% of ZnO content (wt%). The ZnO is self-prepared and the composite obtained via ball milling. The samples are characterized by XRD, SEM, Sieverts-type apparatus. Experiments seem to be carefully done and it is concluded that 2.5 wt%  of ZnO is the best catalyst. Kinetic properties and activation energies are determined. There are some unclear points.Once points raised above are clarified the manuscript can be subject to publication.

Comment 1: Fig. 4: molar mass of three different composition varies, how is it possible that maximum hydrogen content is always the same (~6wt%) and why is decreases with time (@673K) for some samples?

Answer: We thank the reviewer for this comment. Through careful observation, it can be seen that the maximum hydrogen content decreases with the increase of ZnO content, but the decrease is not particularly obvious, which is well demonstrated by TPD test. This mag be caused by the fact that the ZnO can not storage hydrogen and thus the hydrogen capacity is decreased. When the sample absorbs hydrogen at high temperature, the sample will also undergo hydrogen desorption process. However, at this temperature, the rate of hydrogen absorption is much higher than the rate of hydrogen desorption, and the presence of catalyst also accelerates the process of hydrogen absorption. The hydrogen absorption process belongs to the gas-solid reaction, which will eventually reach the equilibrium state. Due to the existence of catalyst, the hydrogen absorption rate is too fast, leading to the increase of hydrogen absorption, but it is not in the equilibrium state at this time. After a period of time, the equilibrium state is reached, resulting in the decrease of the maximum hydrogen absorption. The related discussion has been added in the revised manuscript in line 186.

Comment 2: There is ZnO and after the first re-hydrogenation ZnO is not present (Fig. 3), MgO seems to be present there. In Fig. 8 they claim no MgO. The question is-where is oxygen? Or what means the statement “It can be see that after the addition of ZnO, the catalyst not decomposes.”?

Answer: We thank the reviewer for this comment. Figure 3a shows the samples of Mg-ZnO composites after ball milling. It can be seen that ZnO diffraction peaks are enhanced with the increase of ZnO content. And the elemental mapping in Figure 8 indicates that Mg is uniformly while the places of Zn element concentration is coincided with that of O element, suggesting that the small particles might be ZnO. The XRD pattern shown in Fig. 3a also confirm that ZnO exist in the composite after ball milling, implying that the ZnO should keep stable in the ball milling process. As for MgO, it is inevitable for the formation of MgO for the high activity of the powder after ball milling, therefore, the O element in Fig. 8 is also contributed from MgO to some extent. The related discussion has been added in the revised manuscript in the discussion of Fig. 8.

Comment 3: Since ZnO is gone after first hydrogenation/dehydrogenation-what is the catalytic action of their composite for subsequent hydrogenation events?

Answer: We thank the reviewer for this comment. ZnO is gone after first hydrogenation/dehydrogenation, and their composite has a positive catalytic effect on subsequent hydrogenation events. The reasons are as follows: the reported published works proved that the MgZn2 can facilitate the nucleation of MgH2 during the hydrogenation process and. Liu et al. prepared Mg-Zn alloy and characterized its XRD after the hydrogen absorption and desorption process. The results of XRD show that the majority of Mg will generate MgH2 and the rest of Mg combines with Zn to form MgZn2 intermetallics and the Mg-Zn composite shows better performances than that of Mg. (Reference: T. Liu, et al. Int. J. Hydrogen Energy. 36 (2011) 3515-3520.) Moreover, Zhong et al. synthesized Mg (Sn, Zn) solid solution matrix and after hydrogenation, MgZn2 can be in situ generated and the results showed that MgZn2 can promote the nucleation and promote the hydrogen absorption and release process. (H.C. Zhong, et al. Int. J. Hydrogen Energy. 44 (2019) 2926-2933.). Those results can prove that it has a positive catalytic effect on subsequent hydrogenation events. The related discussion has been added in the revised manuscript in line 169.

Comment 4: The language shall be double checked, and strange statements (The Ede, paper exhibite and many others) shall be corrected.

Answer: We thank the reviewer for this comment. We have changed the description of "The Ede" to "The activation energy after dehydrogenation" and "Paper exhibite" to "The study show that". At the same time we carefully checked the manuscript to find the similar error and corrected them in the revised manuscript.

Reviewer 2 Report

The review is in the file.

Author Response

This paper study the influence of the addition of three different percentages of ZnO nanoparticles (2.5, 5 and 7.5 wt% as catalysts on the hydrogenation properties of Mg, especially on the kinetic. The ZnO nanoparticles were prepared chemically and added to Mg by ball milling under very pure Ar atmosphere. ZnO and the ZnO-Mg composites have been characterized by XRD and SEM with chemical maps. The kinetics curves measured upon hydrogen absorption and desorption have been analyzed by a JMAK model. The paper is clearly written, and the conclusions are justified. However, there are a lot of typos and figures that need to be corrected. This work can be recommended for publication in nanomaterial after several corrections detailed below.

Comment 1: The XRD analysis is done by the indexation of the peaks in the diffractogram. The list of peak positions in the text is fastidious and can be omitted. Identification of the peaks in the figure would be better. Ideally the XRD patterns should be refined using a Rietveld analysis which allow to check not only the position but also the intensity and the weight percentage of each phase. The authors should comment on this point.

Answer: We thank the reviewer for this valuable suggestion. It is true that refinement by using the Rittveld analysis was able to check the peak position, and the intensity and weight percentage of each phase. However, this paper mainly studies the phase change of different content of ZnO in the cycle and the catalytic effect of ZnO with different content on MgH2. The study of the content of the phase in the cycle is superficial. In this paper, with the increase of the ZnO content, the intensity of the diffraction peak was increased. In order to better observe the position of the diffraction peak, we amplified the position of fastidious, as shown in revised Figure 3.

Revised Fig. 3

Comment 2: Figure 4: Kinetic of hydrogenation/dehydrogenation. It would be easier to compare the compound if you place all the hydrogenation curves on the first line and all he deshydrogenation curve on the second line, with one sample by column. This means that a and b should be on the same column, a,c,e on the first line and b,d,f, on the second line.

Answer:We thank the reviewer for this valuable suggestion. To make it better for reviewer and readers, we have placed all the hydrogenation curves on the first line and all he dehydrogenation curve on the second line. Moreover, we have modified this section in the revised manuscript in line 178. And please see the revised Figure 4.

Revised Fig. 4

Comment 3: Figure 5: same remark for the organization of the hydrogenation/dehydrogenation curves. The value on the figure are very difficult to read (too small) and what is the precision of the fit: five digits is probably not significant. I ask to prepare a table with the results of the fit of the kinetic curves versus ln t with appropriate the standard deviation. In this table the values of the activation energies for each compound and reported in figure 6 could be given. It will be much easy to read.

Answer: We thank the reviewer for this comment. To make it better for reviewer and readers, we have increased the value on the figure 5. And we have prepared a table with the results of the fit of the kinetic curves ln[-ln(1-α)] vs ln t with appropriate the standard deviation. At the same time, this table has been included the values of the activation energies for each compound. And please see the revised Figure 5 and Table 1.

Revised Fig. 5

Table 1

MgH2-2.5 wt% ZO

The hydrogenation of the ln[-ln(1-α)] vs lnt equation and the standard deviation

The dehydrogenation of the ln[-ln(1-α)] vs lnt equation and the standard deviation

623K: y=0.87964x-2.71778

573K: y=0.60063x-3.2406

523K: y=0.59901x-3.65377

473K:y=3.95905x-30.91898

SD=0.01835

SD=0.00538

SD=0.00183

SD=0.07303

673K: y=0.82937x-4.53888

648K: y=1.3167x-8.72964

623K: y=2.02505x-14.62246

SD=0.01383

SD=0.00268

SD=0.01026

Eab=72.00 KJ/mol

Ede=107.58 KJ/mol

MgH2-5.0 wt% ZnO

623K: y=0.57677x-1.54704

573K: y=0.44032x-1.48661

523K: y=0.72589x-4.54973

473K:y=1.53387x-11.27874

SD=0.01784

SD=0.00950

SD=0.01053

SD=0.03496

673K: y=0.55408x-3.12996

648K: y=0.79183x-4.99526

623K: y=1.10239x-8.0643

SD=0.00978

SD=0.00451

SD=0.01028

Eab=82.66 KJ/mol

Ede=116.35 KJ/mol

MgH2-7.5 wt% ZnO

673K: y=0.5087x-1.54825

623K: y=0.6983x-2.6114

573K: y=0.60072x-3.54235

523K:y=0.65865x-4.83626

SD=0.02310

SD=0.00728

SD=0.00976

SD=0.00971

673K: y=1.11759x-6.31393

648K: y=1.36846x-8.96609

623K: y=2.08886x-15.64828

SD=0.03828

SD=0.04221

SD=0.03016

Eab=88.43 KJ/mol

Ede=128.41 KJ/mol

Comment 4: Figure 7: the numerotation on each image ((a), (b) etc ) is written in black on a dark background, and is almost not visible. I propose to write them is white as done for scale.

Answer: We thank the reviewer for this comment. To make it better for reviewer and readers, we have changed the numerotation on each image in Figure 7 to white. And please see the revised Figure 7.

Revised Figure 7

Comment 5: Many typos should be corrected, and some sentences are not clearly written.

Page 1, line 41 a dot is missing

Page 2, line 55-56 “in the cycle to decrease”: the verb is missing

Page 3, line 98: what is the meaning of 0 Mpa hydrogen pressure? Under vacuum?

Page 4 line 123: check “sates”?

Page 6 line 192: precise what means JMAK?

Page 6: line 177 : Arrhenius (u is missing)

Page 6: line 177-192: many Ede, Eab are not well written, probably ab, de should be in subscript.

Line 199-201: the hydrogenation temperatures decreased by …, compared to what?

line 208, idem for MgH2

line 212 : shows that

page 8, line 253: ZnO “not decomposes” replace by “is not decomposed”

page 9: line 258” Compare” should be replaced by “compared”

Answer: We thank the reviewer for this valuable suggestions. In the light of the above typos and unclear sentences, we have corrected it one by one and expressed the meaning of the sentence clearly. And we carefully checked the manuscript to find the similar error and corrected them in the revised manuscript.

Round 2

Reviewer 1 Report

The authors might want to:

Check carefully the first two sentences of the Abstract.

Clarify where the home is in the sentence “researchers at home and abroad…”

Check the clarity of the statement “and under vacuum hydrogen pressure.”

Check the clarity of the statement “the quasi certainty of the results”

Check carefully the  statement “This mag be caused by”

etc.

Author Response

Comment 1: Check carefully the first two sentences of the Abstract.

Answer: We thank the reviewer for this comment. We have changed the description of “The ZnO nanoparticles of spheroidal structure via filtration and calcination methods. And the different amount of prepared ZnO were added to MgH2 by ball milling” to “The ZnO nanoparticle in spherical-like structure were synthesized via filtration and calcination methods. And different amount of ZnO nanoparticles were added to MgH2 by ball milling” and we carefully checked the manuscript to find the similar error and corrected them in the revised manuscript.

Comment 2: Clarify where the home is in the sentence “researchers at home and abroad…”

Answer: We thank the reviewer for this comment. We have changed the description of “researchers at home and abroad” to “domestic and foreign scholars” and we carefully checked the manuscript to find the similar error and corrected them in the revised manuscript.

Comment 3: Check the clarity of the statement “and under vacuum hydrogen pressure”

Answer: We thank the reviewer for this comment. We have changed the description of “and under vacuum hydrogen pressure” to “and under vacuum (≤0.1 Pa)” and we carefully checked the manuscript to avoid the similar error.

Comment 4: Check the clarity of the statement “the quasi certainty of the results”

Answer: We thank the reviewer for this comment. We have changed the description of “the quasi certainty of the results” to “the accuracy of the results of the fitting curve of ln[-ln(1-ɑ)] vs lnt at different temperatures were further verified” and we carefully checked the manuscript to avoid the similar error.

Comment 5:Check carefully the statement “This mag be caused by”etc.

We thank the reviewer for this comment. We have changed the description of “This mag be caused by the fact that the ZnO can not storage hydrogen and thus the hydrogen capacity is decreased” to “This is caused by the inability of ZnO to store hydrogen, thus reducing the hydrogen capacity of MgH2-xZnO composites” and we carefully checked the manuscript to avoid the similar error.
